# Deep Attentive Belief Propagation: Integrating Reasoning and Learning for Solving Constraint Optimization Problems

**Yanchen Deng**
School of Computer Science and Engineering
Nanyang Technological University
ycdeng@ntu.edu.sg

**Shufeng Kong** *
School of Software Engineering
Sun Yat-sen University
kongshf@mail.sysu.edu.cn

**Caihua Liu**
School of Artificial Intelligence
Guilin University of Electronic Technology
caihua.liu@alumni.uts.edu.au

**Bo An**
School of Computer Science and Engineering
Nanyang Technological University
boan@ntu.edu.sg

## Abstract

Belief Propagation (BP) is an important message-passing algorithm for various reasoning tasks over graphical models, including solving the Constraint Optimization Problems (COPs). It has been shown that BP can achieve state-of-the-art performance on various benchmarks by mixing old and new messages before sending the new one, i.e., *damping*. However, existing methods of tuning a *static* damping factor for BP not only are laborious but also harm their performance. Moreover, existing BP algorithms treat each variable node's neighbors equally when composing a new message, which also limits their exploration ability. To address these issues, we seamlessly integrate BP, Gated Recurrent Units (GRUs), and Graph Attention Networks (GATs) within the message-passing framework to reason about *dynamic* weights and damping factors for composing new BP messages. Our model, Deep Attentive Belief Propagation (DABP), takes the factor graph and the BP messages in each iteration as the input and infers the optimal weights and damping factors through GRUs and GATs, followed by a multi-head attention layer. Furthermore, unlike existing neural-based BP variants, we propose a novel *self-supervised* learning algorithm for DABP with a smoothed solution cost, which does not require expensive training labels and also avoids the common out-of-distribution issue through efficient online learning. Extensive experiments show that our model significantly outperforms state-of-the-art baselines.

## 1 Introduction

Belief Propagation (BP) [27, 36] is an important message-passing algorithm for various reasoning tasks over graphical models, e.g., computing partition function of a Markov random field [50], estimating the marginal distribution of a given set of random variables [1], and decoding LPDC codes [32]. Constraint Optimization Problems (COPs) [33, 41] are a general mathematical paradigm for modeling many real-world problems like transportation, supply chain, energy, finance, and scheduling [6, 24, 25, 41]. When solving COPs, BP, also known as Min-sum message passing [16], seeks to find a cost-optimal solution by propagating cost information over the corresponding factor graph.

---

*Correspondence to: Shufeng Kong <kongshf@mail.sysu.edu.cn>

36th Conference on Neural Information Processing Systems (NeurIPS 2022).

It is known that vanilla BP does not guarantee convergence on a factor graph containing loops and thus, it often explores low-quality solutions on COPs with cyclic factor graphs due to excessive loopy propagation. Therefore, considerable research efforts [7, 9, 38–40, 54] have been devoted to tackling the convergence issue of loopy BP. Among them, Damped BP (DBP)[1] [9] has drawn significant attention recently. By mixing the new message composed in each iteration with the old message composed in the previous iteration, i.e., damping, it has been shown that DBP with a suitable damping factor will often converge and achieve state-of-the-art performance in practice. In fact, damping can be considered as a degree of freedom to balance exploration and exploitation in BP [9].

Due to the significant impact of damping on BP [53], a damping factor is often treated as a hyperparameter and requires laborious case-by-case tuning. Besides, existing works often use a homogeneous and static damping factor for all variable-nodes which limits DBP's capability. Moreover, DBP simply treats each variable-node's neighbors equally when composing a new message, which fails to explore optimal composition strategy. On the other hand, Deep Neural Networks (DNNs) have been applied to boost the performance of BP [28, 43, 51]. However, existing DNN-based BP variants often require supervised learning and expensive training labels and thus, they cannot be directly applied to solve COPs when training labels are not available. They also manipulate or simulate the BP messages with some black-box DNN components, which destroys the semantic of a BP message being the weighted sum of cost functions' marginalizations and thus, the variable decision-making rule of BP for COPs is no longer applied (i.e., selecting the assignments with the minimum beliefs).

To address the above issues, we propose the *first self-supervised* DNN-based BP for solving COPs by seamlessly integrating BP, Gated Recurrent Units (GRUs) [8], and Graph Attention Networks (GATs) [46] within the massage-passing framework to reason about *dynamic* weights and damping factors for composing new BP messages. Specifically, we make the following key contributions:

- We extend DBP by allowing dynamic damping factors and different neighbor weights for each variable-node. Consequently, we have more fine-grained control for composing new messages to trigger effective exploration without altering the semantic of BP messages.

- We infer the optimal damping factors and neighbor weights for each iteration automatically by seamlessly integrating BP and DNNs. Our model, Deep Attentive Belief Propagation (DABP)[2], firstly embeds respectively the factor graph and BP messages with GATs and GRUs, and infers damping factors and neighbor weights through a multi-head attention [2, 44] layer; Then, BP runs with those updated factors and weights to compose and propagate new messages, and so on. Note that integrating GRUs can capture the dynamics of BP messages and avoid the gradient vanishing/exploding issue in long BP reasoning sequences.

- We further equip our DABP with a novel self-supervised learning loss which is a smoothed surrogate for the cost under BP decision-making rule. Therefore, unlike existing DNN-based BP variants, we do not require expensive training labels which are often not available in practice, and also our model can avoid out-of-distribution issue with efficient online learning.

- We conduct extensive experimental evaluations on four standard benchmarks. The results show that our DABP achieves significantly higher convergence rate, where it successfully converges on average 96.25% instances given 1000 iterations, and it also outperforms the state-of-the-art baselines by considerable margins.

## 2 Related Work

Although COP is NP-hard in general [41], many algorithms have been proposed to solve it in the last several decades, such as BP, local search [21], branch and bound [30], and bucket elimination [11, 12, 18]. Among them, Min-sum BP is an effective approximate algorithm for solving COPs and has been successfully applied to many real-world scenarios [16, 22, 31]. However, vanilla BP offers no convergence guarantee and usually returns low-quality solutions on problems with cyclic factor graphs. Rogers et al. proposed to ensure convergence and have a lower bound on solution quality by relaxing the problem, i.e., removing edges to have acyclic factor graphs [38] through minimization marginalization. Further, Rollon et al. improved the bounding scheme by

---

[1]The method is referred as "Damped Max-sum" in [9]. We use Damped BP for a coherent presentation.
[2]The implementation is publicly available at https://github.com/ycdeng-ntu/DABP

considering both minimization and maximization marginalization [39] and decomposing high-arity factors into unary functions [40]. On the other hand, Zivan et al. proposed to avoid loopy message propagation by strictly controlling the message-passing direction [54]. They also introduced the Value Propagation (VP) scheme to enforce exploitation. To balance exploration and exploitation, a set of non-consecutive VP strategies are proposed [7] as well. Recently, Damped BP (DBP) [9] has been proposed to enhance convergence as well as solution quality in loopy BP by mixing the new messages composed in each iteration with the old messages in previous iterations with a suitable damping factor. However, selecting a good damping factor often requires expertise and laborious case-by-case tuning [53], and DBP uses a static damping factor and treats each node's neighbors equally when composing a new message, which limits its exploration. Our DABP overcomes the aforementioned issues by learning to attend neighbors with dynamic damping factors and neighbor weights when composing new BP messages. Obviously, DABP generalizes DBP since DBP can be considered as a static version of DABP. Moreover, equipped with a self-supervised learning algorithm, our DABP is able to infer the optimal damping factors and weights to enhance the performance of DBP.

On the other hand, LP-based methods solve a COP by leveraging message-passing to solve a Linear Program (LP) relaxation of the combinatorial problem. MPLP [19] is a simple BP variant that solves the convex-dual of the LP relaxation via block gradient descent. Unfortunately, the global convergence is not always guaranteed since coordinate descent may get stuck in suboptimal points. Tree-reweighted belief propagation (TRBP) [48] and Fractional Belief Propagation [49] consider respectively a linear combination of free energies defined on spanning trees of the factor graph and a fractional free energy that generalizes the well-known Bethe free energy. Finally, Norm-Product [20] provides a uniform framework that generalizes Max-product, Sum-product and their TRBP counterpart. Unlike these methods, our DABP directly solve the problem by learning optimal hyperparameters for DBP, and does not require any form of LP relaxation which may either induce intractable number of constraints or only offer a lower bound [29].

Recently, there has been increasing interest in applying DNNs to boost the performance of BP. The most relevant works including: NEBP [43] learns to refine BP messages with some black-box DNN components in each iteration and is applied to solve the error correction decoding task; In the same vein, BPNN [28] uses damping layers to modify factor-to-variable messages and is applied to estimate the partition function of Probabilistic Graphical Models (PGMs); FGNN [51] generalizes Graph Neural Networks (GNNs) to represent Max-Product BP and is applied to find approximate Maximum A Posteriori (MAP) assignment of a PGM. However, all these methods require supervised learning and cannot be directly applied to solve COPs when training labels are not available; They also destroy the semantic of a BP message being the weighted sum of cost functions' marginalizations. Contrarily, we only learn optimal dynamic factors and neighbor weights with DNNs to generalize DBP, which preserves the semantic of the BP messages and also enables us using a self-supervised manner to train our model. Therefore, our model does not require expensive training labels which are often not available in practice due to the heavy computation of exact solvers; and also avoids out-of-distribution issue and enjoys a nice anytime property [52] through efficient online learning.

Our method is also closely related to using DNNs to solve combinatorial problems [5, 14, 26, 34, 47]. In the context of COP solving, Neural RM-RRS [15] uses Multi-layer Perceptrons (MLPs) to parameterize high-dimensional regret tables produced by context-based regret-matching. Similarly, Deep Bucket Elimination (DBE) [37] uses MLPs to approximate the large bucket functions [11]. Recently, Deng et al. [14] proposed a pretrained cost model which predicts the optimal cost of a given partially instantiated COP. The predicted cost is then used to construct heuristics for various COP algorithms such as Large Neighborhood Search (LNS) [21] and backtracking search. These methods either replace some heavy computation components of COP algorithms by a fast DNN approximation, or learn effective heuristics (or hyperparameters) to boost the performance of COP algorithms, and our method falls into the second category.

## 3   Backgrounds

In this section, we review the backgrounds of COPs, factor graphs, and Min-sum BP.

## 3.1 Constraint Optimization Problems and Factor Graph

A Constraint Optimization Problem (COP) [33] is defined by a triplet $\langle X, D, F \rangle$, which corresponds to the set of variables, domains, and constraint functions, respectively. Each variable $x_i \in X$ is associated with a finite domain $D_i \in D$. Each constraint function $f_\ell \in F$ with scope $scp(f_\ell) \subseteq X$ specifies the cost for each possible assignment of the variables in its scope. The objective is to find a solution $\tau^* = (\tau_1^*, \ldots, \tau_{|X|}^*) \in \Pi_{x_i \in X} D_i$ such that the total cost is minimized:

$$\tau^* = \underset{\tau \in \Pi_{x_i \in X} D_i}{\arg\min} \sum_{f_\ell \in F} f_\ell(\tau|_{scp(f_\ell)}), \tag{1}$$

where $\tau|_{scp(f_\ell)}$ is the projection of $\tau$ on $scp(f_\ell)$. A COP can be represented by a factor graph which is a bipartite graph consisting of variable-nodes and function-nodes. Variable-nodes correspond to the variables, and function-nodes correspond to the constraint functions in a COP. An edge between a variable-node and a function-node is established if the variable belongs to the scope of the function.

## 3.2 Min-sum Belief Propagation

Min-sum Belief Propagation (Min-sum BP) [16] is an important algorithm for COPs which performs message-passing on factor graphs. More specifically, the message sent from a variable-node $x_i$ to its neighbor $f_\ell$ in iteration $t$ is a function $\mu_{x_i \to f_\ell}^t : D_i \to \mathbb{R}$ computed by

$$\mu_{x_i \to f_\ell}^t = \sum_{f_m \in \mathcal{N}_i \setminus \{f_\ell\}} \mu_{f_m \to x_i}^{t-1}, \tag{2}$$

where $\mathcal{N}_i$ is the neighbors of $x_i$ in the factor graph, and $\mu_{f_m \to x_i}^{t-1} : D_i \to \mathbb{R}$ is the message sent from $f_m$ to $x_i$ in the previous iteration. Similarly, $f_\ell$ computes a message for its neighbor $x_i$ by

$$\mu_{f_\ell \to x_i}^t = \min_{\mathcal{N}_\ell \setminus \{x_i\}} \left( f_\ell + \sum_{x_j \in \mathcal{N}_\ell \setminus \{x_i\}} \mu_{x_j \to f_\ell}^{t-1} \right). \tag{3}$$

Finally, variable-node $x_i$ makes a decision by choosing a value with minimum belief cost:

$$\tau_i^t = \underset{\tau_i \in D_i}{\arg\min} \sum_{f_\ell \in \mathcal{N}_i} \mu_{f_\ell \to x_i}^t(\tau_i), \tag{4}$$

where $\mu_{f_\ell \to x_i}^t(\tau_i)$ is the belief cost of assignment $\tau_i$ in the message $\mu_{f_\ell \to x_i}^t$.

However, vanilla Min-sum BP usually suffers from non-convergence and explores low-quality solutions on cyclic COPs due to excessive loopy propagation. Damped BP (DBP) [9] attempts to alleviate the issues by damping the messages sent from variable-nodes to function-nodes. Formally,

$$\mu_{x_i \to f_\ell}^t = \lambda \mu_{x_i \to f_\ell}^{t-1} + (1 - \lambda) \sum_{f_m \in \mathcal{N}_i \setminus \{f_\ell\}} \mu_{f_m \to x_i}^{t-1}. \tag{5}$$

By choosing a suitable damping factor $\lambda \in (0, 1]$, DBP can drastically increase the chance of convergence as well as the performance of vanilla Min-sum BP.

# 4 Deep Attentive Belief Propagation

Although message damping has proved to be an effective technique to improve the performance of BP on COPs [9], it still suffers from various issues. First, selecting a damping factor is largely accomplished by empirical tuning, which could be laborious and the number of trials is limited by computational resources and/or runtime. Second, using a homogeneous and static damping factor for all variable-nodes may limit DBP's capability. Finally, DBP simply treats each variable-node's neighbors equally when composing a new message, which fails to explore optimal message composition strategy.

We address the above limitations by allowing dynamic damping factors and different neighbor weights for each variable-node. More specifically, in our framework a variable-node $x_i$ computes a message to function-node $f_\ell$ at iteration $t$ by

$$\mu_{x_i \to f_\ell}^t = \lambda_{i \to \ell}^t \mu_{x_i \to f_\ell}^{t-1} + (1 - \lambda_{i \to \ell}^t)(|\mathcal{N}_i| - 1) \sum_{f_m \in \mathcal{N}_i \setminus \{f_\ell\}} w_{m \to i}^t(\ell) \mu_{f_m \to x_i}^{t-1}, \tag{6}$$

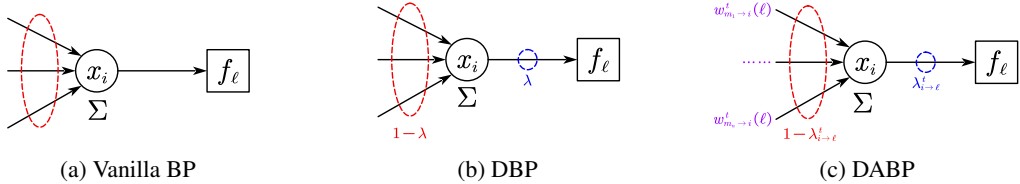



(a) Vanilla BP          (b) DBP          (c) DABP



Figure 1: Comparison of BP variants. Obviously, our DABP generalizes both vanilla BP and DBP.

where $\lambda_{i\to\ell}^t \in [0,1]$ and $w_{m\to i}^t(\ell) \in [0,1]$ are learnable damping factor and neighbor weights for the message from $x_i$ to $f_\ell$ at iteration $t$, respectively. Particularly, we require $\sum_{f_m \in \mathcal{N}_i \setminus \{f_\ell\}} w_{m\to i}^t(\ell) = 1$ and use $|\mathcal{N}_i| - 1$ to rescale the weighted sum of the messages from neighbors. Consequently, we have more fine-grained control for composing new messages to trigger effective exploration without altering the semantic of BP messages. Fig. 1 compares three different BP variants.

A key challenge of our framework is to determine the values of a large number of *time-varying* damping factors and neighbor weights (cf. Eq. (6)). Given a factor graph with $|X|$ variables and maximum degree of $d$, running for $T$ iterations would require to select $O(T|X|d^2)$ hyperparameters from the continuous range of $[0,1]$, which obviously cannot be done by the traditional trial-based hyperparameter tuning methods (e.g., grid search, evolutionary optimization).

Therefore, we propose to automatically infer the optimal hyperparameters for Eq. (6) in each iteration by seamlessly integrating BP and DNNs within the parallel message-passing framework. Our model, Deep Attentive Belief Propagation (DABP), directly outputs the optimal damping factors and weights by learning to attend neighbors given the factor graph and the BP messages in each iteration:

$$w^t, \lambda^t = \mathcal{M}_\theta\left(FG, \mu_{x\to f}^{1:t-1}, \mu_{f\to x}^{1:t-1}\right), \tag{7}$$

where $FG$ is the factor graph, $\mu_{x\to f}^{1:t-1}$ and $\mu_{f\to x}^{1:t-1}$ are the BP messages from variable-nodes to function-nodes and from function-nodes to variable-nodes in the previous iterations, respectively.

However, implementing such DNN-based model requires to perform inference along with BP message-passing iterations (cf. Eq. (7)), which could incur a severe gradient vanishing or gradient exploding problem in a long reasoning sequence. Moreover, obtaining optimal training labels can be extremely expensive for COPs due to the heavy computation of exact solvers, making it impracticable to perform supervised model training on large instances. In Sect. 4.1, we address the first problem by introducing GRUs as a part of our encoder; while we design a novel self-supervised loss function and an efficient online learning framework in Sect. 4.2 to address the second issue.

## 4.1 Model Architecture

Fig. 2 gives the overall architecture of our DABP. It consists of an encoder to embed the factor graph and the BP messages, and an attention module to infer the dynamic optimal hyperparameters.

**Encoder.** The purpose of our encoder is to embed the factor graph given a sequence of BP messages (cf. Fig. 2(a-d)). Therefore, to capture the BP message dynamics and avoid gradient vanishing/exploding issue, we first use two GRUs to embed the BP messages from variable-nodes to function-nodes and the ones from function-nodes to variable-nodes into $q$-dimensional vectors, respectively (cf. Fig. 2(b)):

$$h_{x\to f}^t = GRU_{\phi_1}(h_{x\to f}^{t-1}, \mu_{x\to f}^{t-1}), \quad h_{f\to x}^t = GRU_{\phi_2}(h_{f\to x}^{t-1}, \mu_{f\to x}^{t-1}), \tag{8}$$

where $h_{x\to f}^t, h_{f\to x}^t \in \mathbb{R}^{r\times q}$ are the message hiddens for iteration $t$ and $r$ is the number of edges in the factor graph.

To consider both the topology of a factor graph and the message dynamics, we insert a set of *message nodes* that carry the message hiddens (i.e., the brown and green dots in Fig. 2(c)) and make the graph directed to reflect the message-passing directions. Then the augmented factor graph is embedded through $G$ layers of GAT (cf. Fig. 2(d)). Formally, in the $g \in \{1, \ldots, G\}$-th layer $GAT_{\psi_g}$, we

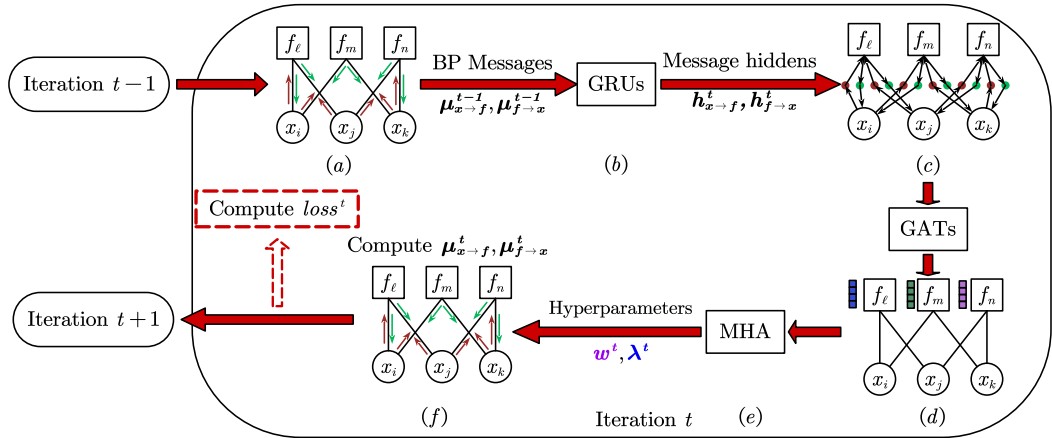

Figure 2: The overall architecture of DABP: (a) The BP messages from iteration $t-1$; (b) To update the message hidden vectors with the messages from iteration $t-1$. Note that these message hidden vectors capture the dynamics of a sequence of BP messages; (c) To insert message nodes with the hidden vectors into the factor graph; (d) To embed the augmented factor graph; (e) To infer optimal hyperparameters by using a multi-head attention layer (MHA); and (f) To compose and propagate new messages with updated hyperparameters and compute training loss in iteration $t$.

compute the embedding $e_i^{t,(g)}$ for node $i$ according to

$$e_i^{t,(g)} = \text{LeakyReLU}\left(\sum_{j \in \mathcal{N}_i} \alpha_{ij}^{(g)} \boldsymbol{W}^{(g)} e_j^{t,(g-1)}\right), \quad \alpha_{ij}^{(g)} = \frac{\exp(s_{ij}^{(g)})}{\sum_{j' \in \mathcal{N}_i} \exp(s_{ij'}^{(g)})}, \quad (9)$$

where $\boldsymbol{W}^{(g)} \in \psi_g$ is a learnable matrix, $s_{ij}^{(g)} = a^{(g)}(\boldsymbol{W}^{(g)} e_i^{t,(g-1)}, \boldsymbol{W}^{(g)} e_j^{t,(g-1)})$ is the attention score between nodes $i$ and $j$, and $a^{(g)}$ is a single-layer feed-forward neural network.

**Attention Module.** Given the embedding $e_\ell^{t,(G)}$ for each $f_\ell \in F$, we update the damping factors and neighbor weights by using a multi-head attention (MHA) layer [44] (cf. Fig. 2(e)), where queries and keys are the embeddings of the target and corresponding neighboring function-nodes, respectively. Specifically, for each variable-node $x_i$ and a target function-node $f_\ell \in \mathcal{N}_i$, we first compute an attention score for each neighboring function-node $f_m \in \mathcal{N}_i$ in $k \in \{1, \ldots, K\}$-th head $Att_{\zeta_k}$ by:

$$a_m^{t,(k)}(\ell) = \sigma\left(\boldsymbol{W_1^{(k)}}\left[\boldsymbol{W_2^{(k)}} e_\ell^{t,(G)} || \boldsymbol{W_3^{(k)}} e_m^{t,(G)}\right]\right), \quad (10)$$

where $\sigma$ is the Sigmoid function, $||$ is the concatenation operation and $\boldsymbol{W_1^{(k)}}, \boldsymbol{W_2^{(k)}}, \boldsymbol{W_3^{(k)}} \in \zeta_k$ are learnable matrices, respectively. Then the attention scores are normalized to compute neighbor weights respect to target function-node $f_\ell$:

$$w_{m \to i}^{t,(k)}(\ell) = \frac{\exp\left(a_m^{t,(k)}(\ell)\right)}{\sum_{f_{m'} \in \mathcal{N}_i \setminus \{f_\ell\}} \exp\left(a_{m'}^{t,(k)}(\ell)\right)}, \quad \forall f_m \in \mathcal{N}_i \setminus \{f_\ell\}. \quad (11)$$

Similarly, the damping factor is computed by normalizing the attention score of $f_\ell$ w.r.t. the mean attention score of the remaining neighboring function-nodes:

$$\lambda_{i \to \ell}^{t,(k)} = \frac{\exp\left(a_\ell^{t,(k)}(\ell)\right)}{\exp\left(a_\ell^{t,(k)}(\ell)\right) + \exp\left(\frac{1}{|\mathcal{N}_i|-1} \sum_{f_{m'} \in \mathcal{N}_i \setminus \{f_\ell\}} a_{m'}^{t,(k)}(\ell)\right)} \quad (12)$$

Finally, both the neighbor weights and damping factors are normalized across different heads, i.e.,

$$\lambda_{i \to \ell}^t = \frac{\sum_{k=1}^K \lambda_{i \to \ell}^{t,(k)}}{K}, \quad w_{m \to i}^t(\ell) = \frac{\sum_{k=1}^K w_{m \to i}^{t,(k)}(\ell)}{K}, \quad \forall f_m \in \mathcal{N}_i \setminus \{f_\ell\}, \quad (13)$$

and are plugged into Eq.(6) to compute new BP messages for iteration $t$ (cf. Fig. 2(f)).

---

**Algorithm 1** A self-supervised online learning algorithm for the DABP model $\mathcal{M}_\theta$

---

**Input:** A COP instance $\mathcal{P} = \langle X, D, F \rangle$, the number of restart $R$, the maximal iteration limit $T_{\max}$, the update interval $T_{\mathrm{upd}} \leq T_{\max}$, and the number of effective iterations $T_{\mathrm{eff}} \in \{1, \ldots, T_{\mathrm{upd}}\}$

1: $\tau^* \leftarrow$ nil; $FG \leftarrow$ factor graph of $\mathcal{P}$
2: **for** $rs = 1, \ldots, R$ **do**
3: $\quad \mu^0_{x_i \to f_\ell} \leftarrow \mathbf{0}, \mu^0_{f_\ell \to x_i} \leftarrow \mathbf{0}, \forall x_i \in X, f_\ell \in \mathcal{N}_i$
4: $\quad$ **for** $t = 1, \ldots, T_{\max}$ **do**
5: $\qquad \boldsymbol{w^t}, \boldsymbol{\lambda^t} \leftarrow \mathcal{M}_\theta \left( FG, \boldsymbol{\mu^{t-1}_{x \to f}}, \boldsymbol{\mu^{t-1}_{f \to x}} \right)$
6: $\qquad$ compute $\mu^t_{x_i \to f_\ell}$ according to Eq. (6) with hyperparameters $\boldsymbol{w^t}, \boldsymbol{\lambda^t}, \forall x_i \in X, f_\ell \in \mathcal{N}_i$
7: $\qquad$ compute $\mu^t_{f_\ell \to x_i}$ according to Eq. (3) $\forall x_i \in X, f_\ell \in \mathcal{N}_i$
8: $\qquad$ compute solution $\tau^t = \langle \tau^t_1, \ldots, \tau^t_{|X|} \rangle$ according to Eq. (4)
9: $\qquad$ **if** $\mathrm{cost}(\tau^*) > \mathrm{cost}(\tau^t)$ **then** $\tau^* \leftarrow \tau^t$
10: $\qquad$ **if** $t \,\%\, T_{\mathrm{upd}} = 0$ **then**
11: $\qquad\quad T^* \leftarrow$ select the best $T_{\mathrm{eff}}$ iterations according to $\mathrm{cost}(\tau^{t'})$, $t - T_{\mathrm{upd}} < t' \leq t$
12: $\qquad\quad$ compute $\mathcal{L}(\theta)$ according to Eq. (16), $\theta \leftarrow$ ADAM$(\theta, \nabla\mathcal{L}(\theta))$
13: $\qquad$ **if** BP messages converge **then break**
$\quad$ **return** $\tau^*$

---

## 4.2 A Novel Self-supervised Learning Algorithm for Solving COPs

Recall that our objective is to find a cost-optimal solution (cf. Eq. (1)), a natural choice of the loss function would be the cost of the solution induced by BP messages in each iteration. Unfortunately, such loss function is not differentiable since variables are greedily assigned via `argmin` (cf. Eq. (4)). Instead, we propose to use a smoothed cost as the surrogate objective. Formally, for each iteration $t$, we consider the following self-supervised objective:

$$loss^t = \sum_{f_\ell \in F} \left( \sum_{\langle \tau_{i_1}, \ldots, \tau_{i_n} \rangle \in \Pi_{x_i \in scp(f_\ell)} D_i} f_\ell(\tau_{i_1}, \ldots, \tau_{i_n}) \prod_{j=1:n} p^t_{i_j}(\tau_{i_j}) \right), \tag{14}$$

where $p^t_i(\tau_i)$ is the probability of $x_i = \tau_i$ under the current belief, i.e.,

$$p^t_i(\tau_i) = \frac{\exp(-b^t_i(\tau_i))}{\sum_{\tau'_i \in D_i} \exp(-b^t_i(\tau'_i))}, \quad b^t_i(\tau_i) = \sum_{f_\ell \in \mathcal{N}_i} \mu^t_{f_\ell \to x_i}(\tau_i). \tag{15}$$

Intuitively, an assignment $\tau_i$ is associated with a high probability if it has a low belief cost $b^t_i(\tau_i)$, which simulates the behavior of `argmin` in Eq. (4) and is in alignment with the objective in Eq. (1).

We now theoretically show the error bound of the smoothed cost. Proof is given in Appendix A.

**Theorem 1.** *Let* $\Delta_\ell = \max_{\langle \tau_{i_1}, \ldots, \tau_{i_n} \rangle} f_\ell(\tau_{i_1}, \ldots, \tau_{i_n}) - \min_{\langle \tau_{i_1}, \ldots, \tau_{i_n} \rangle} f_\ell(\tau_{i_1}, \ldots, \tau_{i_n})$ *be the maximum difference between cost values of each constraint function* $f_\ell$ *and* $\tau^t = \langle \tau^t_1, \ldots, \tau^t_{|X|} \rangle$ *be the assignment induced by Eq. (4). We have*

$$\left| loss^t - \sum_{f_\ell \in F} f_\ell(\tau^t|_{scp(f_\ell)}) \right| \leq \sum_{f_\ell \in F} \Delta_\ell \left( 1 - \prod_{x_i \in scp(f_\ell)} \frac{1}{|D_i|} \right).$$

Given the smoothed objective for each iteration, we define the following loss function:

$$\mathcal{L}(\theta) = \frac{1}{|T^*|} \sum_{t \in T^*} loss^t, \tag{16}$$

where $\theta = \{\phi_1, \phi_2, [\psi_1, \ldots, \psi_G], [\zeta_1, \ldots, \zeta_K]\}$ is the model parameters and $T^*$ is a selected subset of iterations used for optimizing, respectively. Particularly, we rank each iteration $t$ according to the quality of solution $\tau^t$ and select the best $T_{\mathrm{eff}}$ iterations as effective iterations $T^*$. This way, we encourage exploration by considering only good-enough solutions.

Since our model is self-supervised by the smoothed cost, there is no need for costly label generation for each instance and thus, DABP can be directly applied to solve COPs via online learning without

any pre-training procedure. That is, from an initial DABP model $\mathcal{M}_\theta$, we continuously improve it by training on the instance multiple rounds (i.e., *restart*) on the COP to be solved and return the best solution as the result. Therefore, our DABP avoids the out-of-distribution issue and has a nice anytime property [52]. The procedure is detailed in Alg. 1.

Specifically, each round of restart begins with resetting BP messages to zero vectors (line 3), followed by a sequence of message-passing iterations (line 4-13). In each iteration, we query our DABP for the damping factors and neighbor weights (line 5), perform BP message passing according to the updated hyperparameters (line 6-7), and finally update the current best solution (line 8-9). Particularly, we perform model update every $T_{\text{upd}}$ iterations (line 10-12) and thus, we further alleviate the gradient vanishing/exploding issue by restricting the maximum length of reasoning sequence to $T_{\text{upd}}$. Lastly, we early stop each round whenever BP messages converge (line 13) and return the best solution.

## 5    Empirical Evaluations

In this section, we perform extensive empirical studies. We begin with introducing the details of experiments and implementation. Then we analyze the impact of the number of selected effective iterations in Alg. 1. Finally, we show the great superiority of our DABP over the state-of-the-arts.

**Benchmarks & Baselines.**    We consider four types of standard benchmarks in our experiments, i.e., random COPs, scale-free networks, small-world networks, and Weighted Graph Coloring Problems (WGCPs). For random COPs and WGCPs, constraints are randomly established according to graph density $p_1 \in (0, 1]$. For scale-free networks, we use the BA model [4] with parameters $m_0, m_1 \in \mathbb{Z}_+$ to generate constraints. Finally, we generate constraints for small-world networks by using Newman-Watts-Strogatz model [35] with parameters $k \in \mathbb{Z}_+, p \in [0, 1]$.

For baselines, we compare our DABP with the following state-of-the-art COP solvers: (1) DBP with a damping factor of 0.9 and its splitting constraint factor graph version (DBP-SCFG) with a splitting ratio of 0.95 [9]; (2) GAT-PCM-LNS with a destroy probability of 0.2 [14]; (3) Mini-bucket Elimination (MBE) with an $i$-bound of 9 [13], and (4) Toulbar2 with timeout of 1200s [10].

All experiments are conducted on an Intel i9-9820X workstation with GeForce RTX 3090 GPUs and 384GB memory. *We report the best solution cost for each run, and for each experiment we average the results over 100 random problem instances*. Due to the space constraint, we defer the details of benchmarks and baselines to Appendix B.1 and B.2, respectively.

**Implementation.**    Our DABP consists of two GRUs which embeds BP messages into 8-dimensional hidden vectors, followed by $G = 4$ layers of GAT, each of them has 8 output channels and 4 attention-heads. We then infer the optimal damping factors and neighbor weights by using a multi-head attention layer with $K = 4$ attention-heads. Finally, we also adopt the SCFG scheme with the splitting ratio of 0.95 [9]. Our model was implemented with the PyTorch Geometric framework [17] and the model was trained with the Adam optimizer [23] using a learning rate of $10^{-4}$ and a weight decay ratio of $5 \times 10^{-5}$. For each instance, we perform $R = 20$ restarts with iteration limit $T_{\max} = 1000$ and update the model every $T_{\text{upd}} = 20$ iterations.

**Performance Comparison.**    Table 1 compares the performance of different methods on the four standard benchmarks. We observe that, although given a relatively high timeout (i.e., 20 minutes), Toulbar2 still performs poorly on the most test cases. This is not surprising because it systematically explores the whole solution space, which is often infeasible for large problem instances. Also, it highlights the extreme difficulty of obtaining optimal labels with an exact solver for the existing DNN-based BP variants. MBE, on the other hand, performs memory-bounded inference and thus runs much faster than Toulbar. However, MBE fails to find good solutions and is strictly dominated by the other algorithms. GAT-PCM-LNS performs iterative large neighborhood search with a machine-learned repair heuristic and often outperforms DBP. However, GAT-PCM-LNS requires significant longer runtime than DBP since it needs to perform multiple times of model inference in each iteration to re-assign the destroyed variables. With a small modification in a factor graph, DBP-SCFG drastically improves the performance of DBP but it is computationally heavier. That is because the number of function-nodes in DBP-SCFG is doubled due to constraint function splitting.

Table 1: The performance of different methods. Costs are normalized by the number of constraints $|F|$. Gap is the ratio of cost difference w.r.t. the best result. The best results are shown in **bold**.

| | | $|X| = 60$ | | | $|X| = 80$ | | | $|X| = 100$ | |
|---|---|---|---|---|---|---|---|---|---|
| Methods | Cost | Gap | Time | Cost | Gap | Time | Cost | Gap | Time |
| | | | | Random COPs ($p_1 = 0.25$) | | | | | |
| Toulbar2 | 29.26 | 7.88% | 20m | 32.49 | 8.23% | 20m | 34.36 | 7.23% | 20m |
| MBE | 32.04 | 18.11% | 4m2s | 34.85 | 16.10% | 8m38s | 36.76 | 14.72% | 11m58s |
| GAT-PCM-LNS | 28.00 | 3.21% | 5m27s | 30.80 | 2.59% | 12m55s | 32.78 | 2.31% | 24m47s |
| DBP | 27.86 | 2.72% | 1m11s | 30.79 | 2.58% | 2m25s | 33.06 | 3.17% | 4m18s |
| DBP-SCFG | 27.60 | 1.77% | **52s** | 30.50 | 1.60% | 2m6s | 32.45 | 1.28% | 3m59s |
| DABP ($R = 5$) | 27.19 | 0.24% | 1m4s | 30.09 | 0.23% | **1m20s** | 32.12 | 0.26% | **1m49s** |
| DABP ($R = 10$) | 27.16 | 0.12% | 2m2s | 30.05 | 0.10% | 2m41s | 32.07 | 0.10% | 3m37s |
| DABP ($R = 20$) | **27.12** | **0.00%** | 4m | **30.02** | **0.00%** | 5m20s | **32.04** | **0.00%** | 7m20s |
| | | | | WGCPs ($p_1 = 0.25$) | | | | | |
| Toulbar2 | **0.18** | **0.00%** | 20m | 1.23 | 41.06% | 20m | 2.11 | 42.09% | 20m |
| MBE | 1.98 | 1028.69% | **0s** | 2.81 | 223.13% | **0s** | 3.43 | 131.05% | **1s** |
| GAT-PCM-LNS | 0.51 | 191.35% | 57s | 1.16 | 33.98% | 2m40s | 1.79 | 20.47% | 5m56s |
| DBP | 1.78 | 913.53% | 29s | 3.03 | 249.36% | 1m4s | 3.79 | 154.99% | 1m55s |
| DBP-SCFG | 0.40 | 130.39% | 30s | 1.04 | 19.77% | 1m59s | 1.69 | 13.97% | 4m29s |
| DABP ($R = 5$) | 0.32 | 80.52% | 1m9s | 0.89 | 2.39% | 2m53s | 1.57 | 5.50% | 5m47s |
| DABP ($R = 10$) | 0.30 | 73.80% | 2m15s | 0.88 | 0.92% | 5m37s | 1.50 | 0.69% | 11m19s |
| DABP ($R = 20$) | 0.29 | 67.12% | 4m26s | **0.87** | **0.00%** | 11m10s | **1.49** | **0.00%** | 22m35s |
| | | | | Scale-free networks ($m_0 = m_1 = 10$) | | | | | |
| Toulbar2 | 31.01 | 7.51% | 20m | 31.70 | 8.07% | 20m | 32.69 | 10.51% | 20m |
| MBE | 33.70 | 16.85% | 4m8s | 34.26 | 16.82% | 5m43s | 34.58 | 16.91% | 7m9s |
| GAT-PCM-LNS | 29.62 | 2.70% | 6m25s | 30.41 | 3.66% | 10m55s | 31.10 | 5.16% | 17m7s |
| DBP | 29.32 | 1.65% | 1m16s | 30.11 | 2.66% | 2m8s | 30.47 | 3.04% | 2m57s |
| DBP-SCFG | 29.27 | 1.46% | 1m18s | 29.76 | 1.47% | 1m51s | 30.01 | 1.48% | 2m27s |
| DABP ($R = 5$) | 28.93 | 0.30% | **1m2s** | 29.43 | 0.32% | **1m15s** | 29.67 | 0.33% | **1m18s** |
| DABP ($R = 10$) | 28.87 | 0.10% | 2m2s | 29.38 | 0.17% | 2m33s | 29.62 | 0.15% | 2m37s |
| DABP ($R = 20$) | **28.84** | **0.00%** | 4m3s | **29.33** | **0.00%** | 5m9s | **29.58** | **0.00%** | 5m12s |
| | | | | Small-world networks ($k = 10, p = 0.3$) | | | | | |
| Toulbar2 | 27.98 | 8.72% | 20m | 28.28 | 10.37% | 20m | 28.36 | 10.60% | 20m |
| MBE | 29.67 | 15.26% | 2m46s | 29.39 | 14.71% | 3m39s | 29.54 | 15.17% | 4m50s |
| GAT-PCM-LNS | 26.75 | 3.93% | 4m | 26.65 | 4.04% | 6m19s | 26.68 | 4.02% | 9m12s |
| DBP | 26.77 | 4.02% | **1m** | 27.03 | 5.52% | **1m23s** | 27.40 | 6.83% | **1m50s** |
| DBP-SCFG | 26.30 | 2.17% | 1m4s | 26.09 | 1.83% | 1m31s | 26.13 | 1.90% | 2m10s |
| DABP ($R = 5$) | 25.84 | 0.39% | 1m15s | 25.70 | 0.33% | 1m31s | 25.73 | 0.34% | 1m56s |
| DABP ($R = 10$) | 25.78 | 0.15% | 2m36s | 25.65 | 0.14% | 3m6s | 25.70 | 0.20% | 3m49s |
| DABP ($R = 20$) | **25.74** | **0.00%** | 5m8s | **25.62** | **0.00%** | 6m20s | **25.65** | **0.00%** | 7m31s |

Our DABP exhibits great superiority in various benchmarks. Specifically, given 5 times of restart, DABP substantially outperforms DBP-SCFG, which is currently the strongest approximate solver for COPs, on all test cases and the gap is widened with the increasing $R$. This demonstrates the merits of our learned dynamic damping factors and neighbor weights over the static and homogeneous counterpart. Importantly, although our DABP requires to perform online learning when solve each instance, our DABP's runtime results are still comparable to DBP-SCFG and also scales up well to large problem instances, thanks to our seamlessly integration of BP and DNNs within the message-passing framework which enables efficient parallel computation on GPUs. In fact, when applied to scale-free networks, DABP ($R = 5$) requires less runtime for all configurations, and has a much lower growing rate ($\sim$23.81%) than DBP-SCFG ($\sim$37.37%). Besides, equipped with the self-supervised loss function for minimizing the smoothed cost, our DABP is able to infer the optimal hyperparameters that foster fast convergence, which also improves our model's efficiency.

**Further Performance Analysis.** We further compare the solution quality of different methods in each iteration and analyze the convergence rate of BP variants on the problem instances with $|X| = 80$[3] in Fig. 3 and Fig. 4, respectively. We omit the cases of DABP with $R = 10$ and $R = 20$ in Fig. 4 since they are similar to the one of DABP ($R = 5$). It can be seen that DBP improves slowly and fails to find good solutions within 1000 iterations, especially on WGCPs. DBP-SCFG

---

[3]The results for the instances with $|X| = 60$ and $|X| = 100$ can be found in Appendix B.4

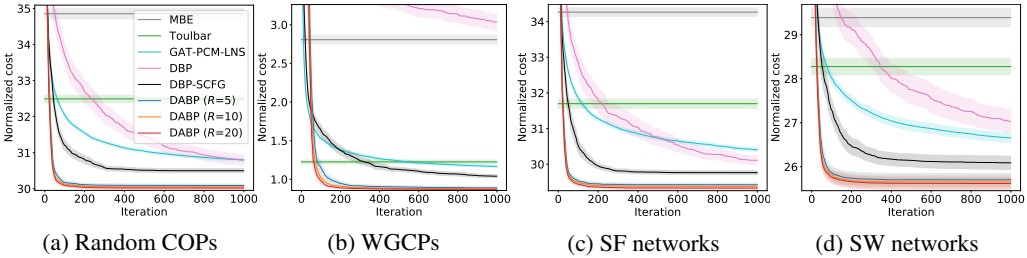

(a) Random COPs    (b) WGCPs    (c) SF networks    (d) SW networks

Figure 3: Solution quality comparison. Shaded areas show the 95% confidence intervals.

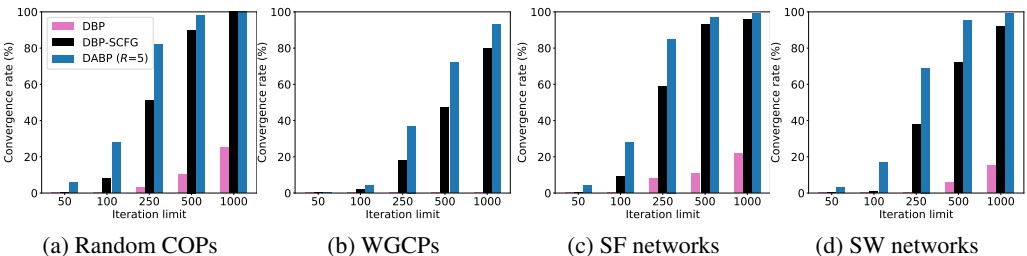

(a) Random COPs    (b) WGCPs    (c) SF networks    (d) SW networks

Figure 4: Convergence rates under different iteration limits.

substantially improves the convergence as well as solution quality of DBP by introducing another degree of freedom through asymmetric function splitting [9]. On the other hand, our DABP allows more flexibility in composing BP messages by reasoning optimal dynamic hyperparameters through seamlessly integrating BP and DNNs, and it also significantly outperforms and converges much faster than both DBP and DBP-SCFG.

## 6 Conclusion

In this paper, we propose DABP, the first self-supervised DNN-based BP for solving COPs. By allowing dynamic damping factors and different neighbor weights for each variable-node, our DABP strictly generalizes the state-of-the-art DBP and has more fine-grained control for composing new messages to trigger effective exploration without altering the semantic of BP messages. Moreover, by seamlessly integrating BP, GRUs and GATs within the massage-passing framework, DABP is able to infer the optimal hyperparameters automatically according to the BP message dynamics. Finally, equipped with a novel self-supervised loss function, our DABP does not require expensive training labels and can avoid out-of-distribution issue with efficient online learning. Extensive experiments confirm the great superiority of our DABP over the state-of-the-art baselines.

Since it relies smoothed costs as the self-supervised loss function, our DABP is dedicated to solve COPs and is not trivial to be adapted for other scenarios. Therefore, in future we will explore more training paradigms to extend our model to solve other important reasoning tasks over graphical models, e.g., computing partition function [50], and finding constrained most probable explanation [42]. It is also interesting to further boost the performance of our model by incorporating popular deep learning techniques such as contrastive learning [3] and transformer [45].

## Acknowledgement

This research is supported by the National Research Foundation, Singapore under its Industry Alignment Fund – Pre-positioning (IAF-PP) Funding Initiative. Any opinions, findings and conclusions or recommendations expressed in this material are those of the author(s) and do not reflect the views of National Research Foundation, Singapore. The work of Caihua Liu is supported and funded by the Humanities and Social Sciences Youth Foundation, Ministry of Education of the People's Republic of China (Grant No.21YJC870009). We also sincerely thank Professor Rina Dechter at University of California, Irvine for her helpful discussions and suggestions.

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
