# A   Proof of Theorem 1

*Proof.*

$$\left| loss^t - \sum_{f_\ell \in F} f_\ell(\tau^t|_{scp(f_\ell)}) \right| = \sum_{f_\ell \in F} \left| f_\ell(\tau^t|_{scp(f_\ell)}) - \sum_{\langle \tau_{i_1}, \dots, \tau_{i_n} \rangle \in \Pi_{x_i \in scp(f_\ell)} D_i} f_\ell(\tau_{i_1}, \dots, \tau_{i_n}) \prod_{j=1:n} p_{i_j}^t(\tau_{i_j}) \right|$$

$$= \sum_{f_\ell \in F} \left| \sum_{\langle \tau_{i_1}, \dots, \tau_{i_n} \rangle \in \Pi_{x_i \in scp(f_\ell)} D_i} \left( f_\ell(\tau^t|_{scp(f_\ell)}) - f_\ell(\tau_{i_1}, \dots, \tau_{i_n}) \right) \prod_{j=1:n} p_{i_j}^t(\tau_{i_j}) \right|$$

$$\leq \sum_{f_\ell \in F} \sum_{\substack{\langle \tau_{i_1}, \dots, \tau_{i_n} \rangle \in \Pi_{x_i \in scp(f_\ell)} D_i \\ \langle \tau_{i_1}, \dots, \tau_{i_n} \rangle \neq \tau^t|_{scp(f_\ell)}}} \Delta_\ell \prod_{j=1:n} p_{i_j}^t(\tau_{i_j})$$

$$= \sum_{f_\ell \in F} \Delta_\ell \left( 1 - \prod_{x_i \in scp(f_\ell)} p_i(\tau_i^t) \right)$$

According to Eq. (4) and Eq. (15), for each $x_i \in X$ it must be the case that $\tau_i^t = \arg\min_{\tau_i \in D_i} b_i^t(\tau_i)$ and hence $\tau_i^t = \arg\max_{\tau_i \in D_i} p_i^t(\tau_i)$. We now show that $p_i^t(\tau_i^t) \geq \frac{1}{|D_i|}$. Assume by contradiction that $p_i^t(\tau_i^t) < \frac{1}{|D_i|}$. Since $\tau_i^t$ has the highest probability,

$$\sum_{\tau_i \in D_i} p_i^t(\tau_i) \leq \sum_{\tau_i \in D_i} p_i^t(\tau_i^t) = |D_i| p_i^t(\tau_i^t) < 1,$$

which contradicts to the fact that $p_i^t$ is a probability distribution over $D_i$. Therefore,

$$\left| loss^t - \sum_{f_\ell \in F} f_\ell(\tau^t|_{scp(f_\ell)}) \right| \leq \sum_{f_\ell \in F} \Delta_\ell \left( 1 - \prod_{x_i \in scp(f_\ell)} p_i(\tau_i^t) \right) \leq \sum_{f_\ell \in F} \Delta_\ell \left( 1 - \prod_{x_i \in scp(f_\ell)} \frac{1}{|D_i|} \right).$$

$\square$

# B   Additional Experimental Details and Results

In this section, we present additional experimental details and results.

## B.1   Benchmarks

We consider four types of benchmarks in our experiments, i.e., random COPs, scale-free networks, small-world networks, and Weighted Graph Coloring Problems (WGCPs). For random COPs and WGCPs, given $|X|$ variables and density of $p_1 \in (0, 1)$, we randomly create a constraint for a pair of variables with probability $p_1$. For scale-free networks, we use the BA model [1] with parameter $m_0$ and $m_1$ to generate constraints: starting from a connected graph with $m_0$ vertices, a new vertex is connected to $m_1$ vertices with a probability which is proportional to the degree of each existing vertex in each iteration. For small-world networks, we generate problem topology according to Newman-Watts-Strogatz model [8]: starting from a ring of $|X|$ vertices, each vertex is connected to its $k$ nearest neighbors, and for each edge underlying the ring with $k$ nearest neighbors, we create a new shortcut by randomly selecting another vertex with a probability $p$. Finally, for each variable and constraint in the benchmarks except WGCPs, we set domain size to 15 and uniformly sample a cost from $[0, 100]$ for each possible assignment combination, respectively. Differently, for WGCPs each variable has a domain of 5 values, and for any pair of constrained variables, we uniformly sample a cost from $[1, 100]$ for an unanimous variable assignment and otherwise, a zero cost.

## B.2   Baselines

Since BPNN [9] and NEBP [7] are proposed for partition function estimation and error correction decoding, respectively, it is not trivial to adopt them for solving COPs. Although FGNN [11] can be applied to solve COPs, all of these existing DNN-based BP variants require expensive supervised

learning and optimal labels which are not available in our experimental settings because we consider large-scale problems and it is not practical to obtain optimal labels using heavy exact solvers. *Therefore, to be fair and practical, we do not consider existing DNN-based BP variants.*

We compare our DABP with the following state-of-the-art COP solvers: (1) DBP with a damping factor of 0.9 and its splitting constraint factor graph version (DBP-SCFG) with a splitting ratio of 0.95 [2]; (2) GAT-PCM-LNS with a destroy probability of 0.2 [5], which is a local search method combining the LNS framework [6, 10] with neural-learned repair heuristics; (3) Mini-bucket Elimination (MBE) with an $i$-bound of 9 [4], which is a memory-bounded inference algorithm; (4) Toulbar2 with timeout of 1200s [3], which is a highly optimized exact solver written in C++. The hyperparameters for DBP and GAT-PCM-LNS are set according to the original papers, while the memory budget for MBE and timeout for Toulbar2 are set based on our computational resources.

All experiments are conducted on an Intel i9-9820X workstation with GeForce RTX 3090 GPUs and 384GB memory. We terminate DBP(-SCFG) and GAT-PCM-LNS whenever convergence or the maximum iteration limit ($T_{\max} = 1000$) reaches. Finally, we report the best solution cost for each run, and for each experiment we average the results over 100 random problem instances.

### B.3 Parameter Tuning

In this subsection, we empirically study the impact of damping factor $\lambda$ on the performance of DBP and DBP-SCFG, and the impact of the number of effective iterations on the performance of our DABP, respectively.

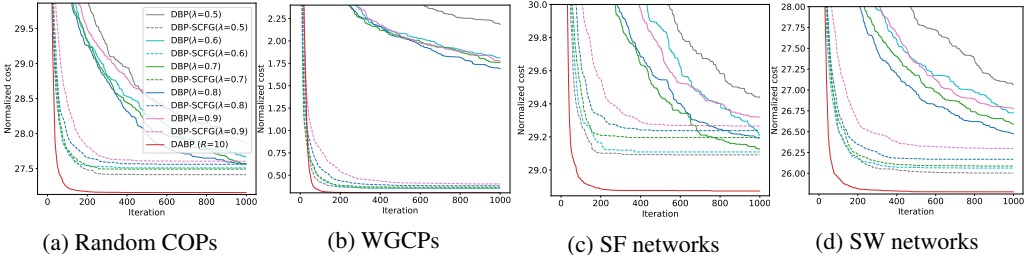

    (a) Random COPs        (b) WGCPs       (c) SF networks      (d) SW networks

Figure 1: Solution quality comparison with different $\lambda$ ($|X| = 60$)

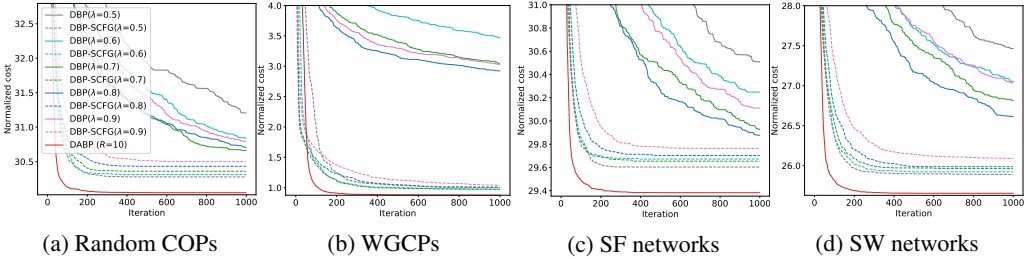

    (a) Random COPs        (b) WGCPs       (c) SF networks      (d) SW networks

Figure 2: Solution quality comparison with different $\lambda$ ($|X| = 80$)

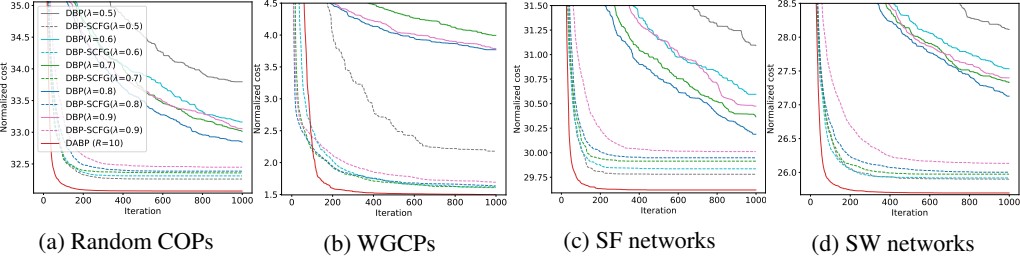

    (a) Random COPs        (b) WGCPs       (c) SF networks      (d) SW networks

Figure 3: Solution quality comparison with different $\lambda$ ($|X| = 100$)

**Impact of Damping Factor** $\lambda$. We vary $\lambda$ from 0.5 to 0.9 with a step size of 0.1, and Fig.1-3 present the solution quality performance of DBP and DBP-SCFG under different damping factors. It can be observed that the performance of DBP varies a lot with different $\lambda$ while DBP-SCFG is relatively less sensitive to the damping factor. In more detail, DBP with a small $\lambda$ (e.g., $\lambda = 0.5$ or $\lambda = 0.6$) trends to produce low-quality solutions, which is due to the fact that a small damping factor usually cannot eliminate the effect of the costs accumulated before the final periodical of the BP process [12]. On the other hand, a large damping factor (e.g., $\lambda = 0.9$) can also lead to poor results, since it requires significantly more iterations to update the beliefs before finding a high-quality solution.

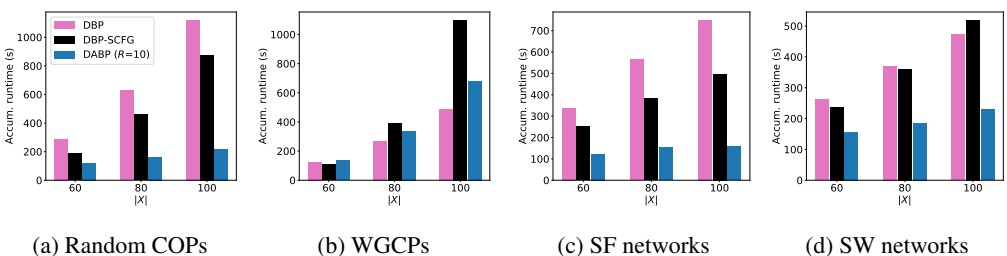

| (a) Random COPs | (b) WGCPs | (c) SF networks | (d) SW networks |

Figure 4: Accumulated runtime comparison

However, it can be extremely tedious and time-consuming to tune the damping factor. Fig. 4 presents the accumulated runtime of tuning damping factor in DBP and DBP-SCFG, and the one of DABP with 10 times of restart. It can be seen that both DBP and DBP-SCFG require significantly higher runtime if we tune the damping factor. In fact, the runtime is generally proportional to the number of damping factors we have attempted. On the other hand, to obtain high-quality solutions one needs to perform extensive tuning (e.g., by using a smaller step size or a wider range), which substantially increases the overall runtime overhead. In contrast, our DABP automatically infers the optimal hyperparameters from the BP messages in the previous iterations, eliminating the need of notoriously laborious tuning procedure and finding better solutions in all test cases with much smaller runtime.

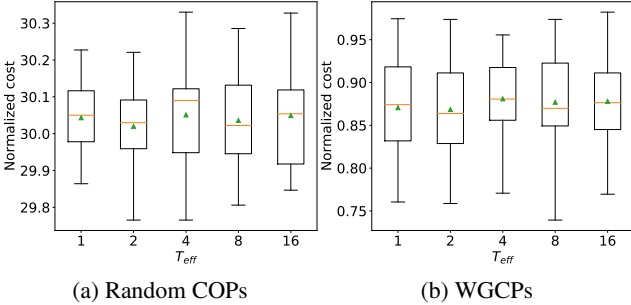

| (a) Random COPs | (b) WGCPs |

Figure 5: Solution quality when varying $T_{\text{eff}}$ (i.e., the number of effective iterations in Alg. 1).

**Impact of the Number of Effective Iterations** $T_{\text{eff}}$. We train our DABP with different $T_{\text{eff}}$ on the Random COPs and WGCPs with 80 variables and $p_1 = 0.25$. Fig. 5 presents the solution quality when varying $T_{\text{eff}}$. It can be seen that a large $T_{\text{eff}}$ usually produces inferior solutions. In such scenario, $T^*$ may contain many suboptimal iterations and their solutions are relatively easy to be improved. Consequently, DABP is more exploitative and thus, it is prone to get trapped in local optima. In contrast, a small $T_{\text{eff}}$ forces DABP to improve only good-enough solutions and encourages exploration. In our experiments, we use $T_{\text{eff}} = 2$ due to its better solution quality.

## B.4 Results on the Instances with 60 and 100 Variables

Fig. 6-9 present solution quality in each iteration and convergence rate on the instances with $|X| = 60$ and $|X| = 100$. Our DABP outperforms the other baselines by considerable margins on a wide range of benchmarks. The only exception happens on WGCPs with 60 variables, where Toulbar2 exhibits the best performance. That is because the variables in WGCPs have a relatively small domain

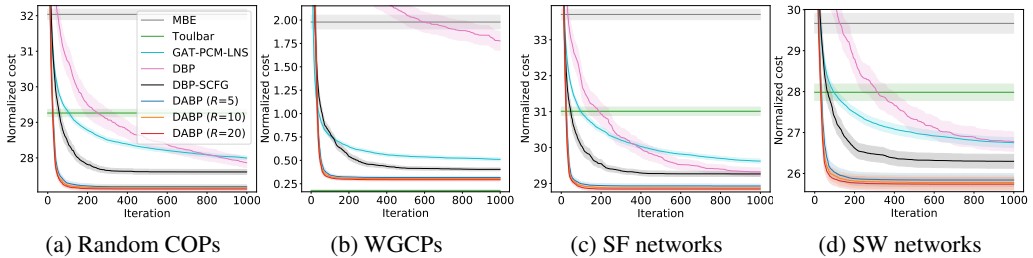

Figure 6: Solution quality comparison ($|X| = 60$)

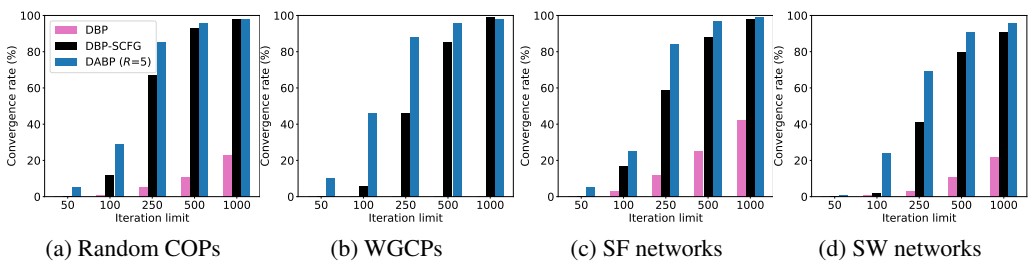

Figure 7: Convergence rates under different iteration limits ($|X| = 60$)

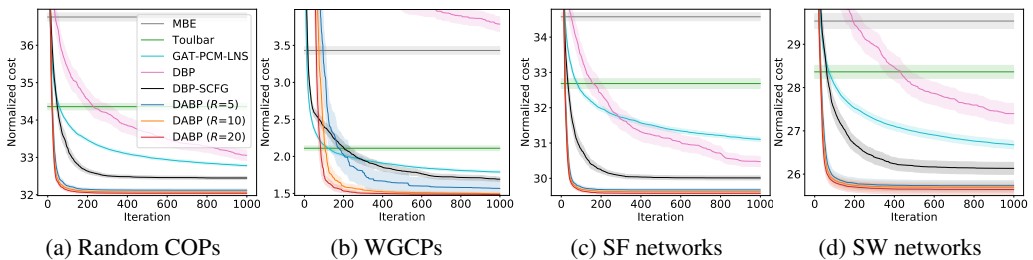

Figure 8: Solution quality comparison ($|X| = 100$)

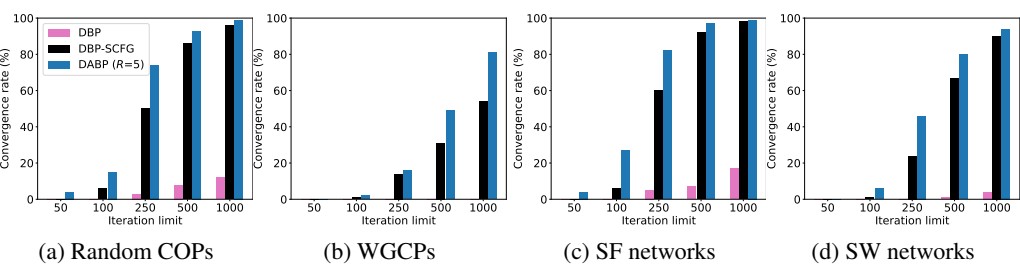

Figure 9: Convergence rates under different iteration limits ($|X| = 100$)

size (i.e., 5) and the constraint functions are highly structured, which allows effective pruning and enumeration for exact methods like Toulbar2. It also can be concluded that our DABP converges much faster than DBP and DBP-SCFG. Notably, when solving WGCPs with 100 variables (cf. Fig. 9(b)), DBP-SCFG has a poor convergence rate and, even worse, DBP entirely fails to converge, while our DABP still achieve convergence on the most of instances, which demonstrates the virtues of our learned dynamic hyperparameters.

## B.5 Ablation Study

To demonstrate the necessity of heterogeneous hyperparameters of Eq. (6), we conduct extensive ablation studies by (1) fixing neighbor weights $w_{m\to i}^t(\ell)$ to $\frac{1}{|N_i|-1}$ and inferring only heterogeneous damping factor $\lambda_{i\to\ell}^t$ (referred as DABP_Heter_$\lambda$), and (2) fixing neighbor weights $w_{m\to i}^t(\ell)$ to $\frac{1}{|N_i|-1}$ and inferring a homogeneous damping factor $\lambda^t$ by averaging all the damping factors computed by Eq. (12) (referred as DABP_Homo_$\lambda$). It is noteworthy that DABP_Heter_$\lambda$ and DABP_Homo_$\lambda$ reduce the number of hyperparameters from $O(T|X|d^2)$ to $O(T|X|d)$ and $O(T)$, respectively.

Fig. 10-12 present the results on solution quality. It can be observed that without heterogeneous neighbor weights DABP_Heter_$\lambda$ often converges to the solutions inferior to the ones found by DABP, given the same number of restarts. DABP_Homo_$\lambda$, on the other hand, performs significantly worse than DABP_Heter_$\lambda$ and DABP, which highlights the merits and necessity of learning heterogeneous hyperparameters in belief propagation for COPs. Nonetheless, equipped with learnable deep neural networks, DABP_Homo_$\lambda$ still substantially outperforms DBP-SCFG which relies solely on a single static damping factor in terms of both solution quality and convergence speed.

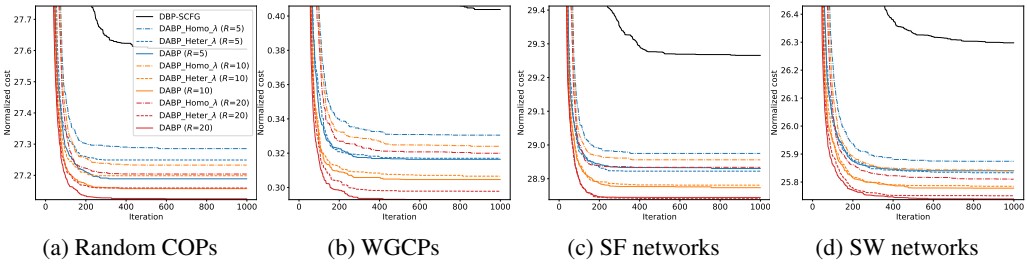

| (a) Random COPs | (b) WGCPs | (c) SF networks | (d) SW networks |

Figure 10: Ablation results on the instances with $|X| = 60$

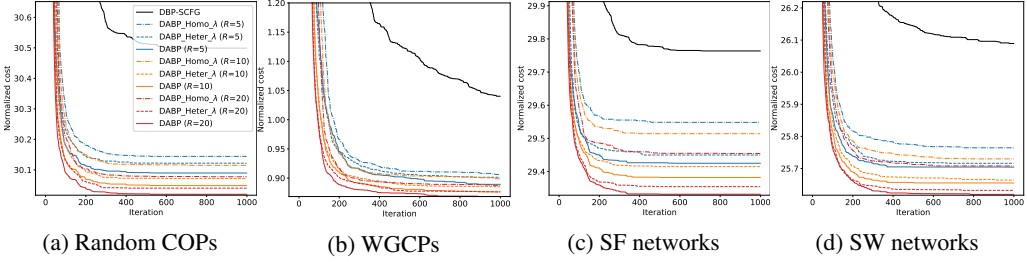

| (a) Random COPs | (b) WGCPs | (c) SF networks | (d) SW networks |

Figure 11: Ablation results on the instances with $|X| = 80$

## B.6 Memory Footprint

Table 1 presents the GPU memory footprint of our DABP. We do not include CPU memory usage since DABP has a similar CPU memory footprint of 4.3GB on all test cases, It can be observed that the GPU memory footprint on random COPs (whose domain size is 15) is similar to the one on WGCPs (whose domain size is 5), which indicates that the GPU memory usage of our DABP is insensitive to the domain size. Besides, on highly-structured problems like scale-free networks and small-world networks, our DABP trends to consume less memory than uniform problems.

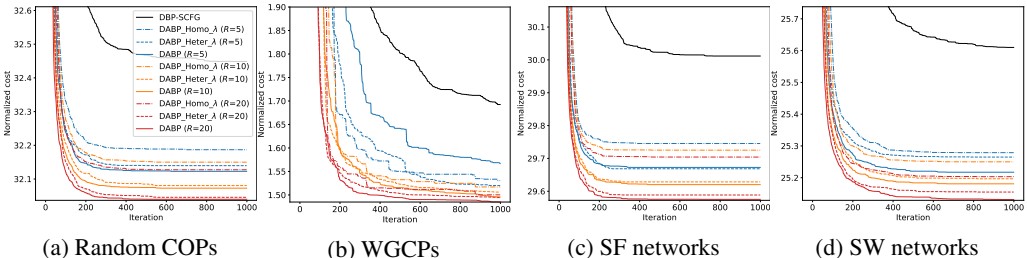

| (a) Random COPs | (b) WGCPs | (c) SF networks | (d) SW networks |

Figure 12: Ablation results on the instances with $|X| = 100$

Table 1: GPU memory footprint of DABP (in GB)

|  | Random COPs | WGCPs | SF Nets | SW Nets |
|---|---|---|---|---|
| $|X| = 60$ | 6.54 | 6.53 | 8.34 | 6.14 |
| $|X| = 80$ | 12.22 | 12.16 | 11.44 | 7.01 |
| $|X| = 100$ | 20.41 | 20.34 | 18.74 | 8.31 |