# OpenReview forum: "Deep Attentive Belief Propagation: Integrating Reasoning and Learning for Solving Constraint Optimization Problems"
_NeurIPS.cc/2022/Conference — NeurIPS 2022 Accept_

### Official Review · Reviewer_rbQp · 2022-07-03

**Rating:** 6
**Confidence:** 3
**Soundness:** 3 good
**Presentation:** 3 good
**Contribution:** 2 fair

**Summary:**

The authors propose a self-supervised DNN Based Belief Propagation solver for Constraint Optimization Problem.
They start from the Dumping Belief Propagation and define an approach to have a dynamic Dumping factor and neighbor weights for the computation of the messages.

**Questions:**

Rows 40-46 --> Please cite properly

Equations --> The formalism used to describe the message could be misleading. Also to be coherent with the summation I suggest to use:
$\mu_{f_m \rightarrow x_i}^{t-1}$ for message sent from factor node $f_m$ to variable node $x_i$. In the same way for the message from the variable node to the factor node.

Equation 3 --> $\mu_{x_j \rightarrow f_l}^{t-1}$ is not introduced

Equation 4 --> what is the meaning of the dependence $\mu_{l \rightarrow i}^{t-1}(\tau_i)$ by $\tau_i$? It is not described.

Usually in the damping BP the weights of the old estimate and of the new estimate sum to 1.

In Eq.6 authors introduce a scaling factor that increases the contribution of the new estimates on the basis of the number of the neighbor nodes. This means that if there are a lot of neighbors the new estimate has higher contribution. The authors should describe better the rationale behind this choice.

Row 217 --> Do the authors mean in term of time? The given motivation is related to the time to reach a solution.

Table 1 - WGCP --> The test on this benchmark requires more discussion. The Toulbar2 outperforms other methods dramatically for |X|=60 and MBE require 0s... please check the results on this benchmark and discuss a little bit more on it.

DBP presents little bit worse results (except for WGCP). Do the authors have used 0.9 as Dumping Factor but, since it is necessary to fine-tune this parameter (as they in stated in section 3), I'd like to see a test with different dumping factor because the selected dumping factor could be not a good choice. The same for DBP-SCFG.

Fig. 5 I suggest to increase the bars' width.

**Limitations:**

Authors described the main limitation, that the extension to other scenarios is not straightforward.
The extension will be study in the future works because deserve more investigations.

**Strengths And Weaknesses:**

The work is well organized and there are extensive experiments with different benchmarks and models.
Some other experiments should be performed to have a fair comparison with simple DBP.

---

> ### Author Response · Authors · 2022-08-02
> **Responses to Reviewer rbQp**
>
> We thank Reviewer rbQp for insightful comments and helpful feedback on our work. We address Reviewer rbQp's suggestions and respond to specific comments below
>
> **1. The notations in BP formalization**
>
> **Response:** We thank the reviewer for suggestion of refining the belief propagation formalization. We have revised the notation and added the missed descriptions.
>
> **2.  Regarding the scaling factor in Eq. (6)**
>
> **Response:** The purpose of introducing the scaling factor $|N_i|-1$ is to keep the magnitude of the messages in our DABP the same as BP and DBP, since we have required the neighbor weights to sum to 1 in Eq. (6). Intuitively, each message from the neighboring function-node contributes 1 unit in the new variable-to-function message in BP (resp. $1-\lambda$ unit in DBP), and thus all the messages from neighboring non-target function-nodes contribute $|N_i|-1$ units in BP (resp. $(1-\lambda)(|N_i|-1)$ units in DBP). With the scaling factor of $|N_i|-1$, each message in our DABP contributes $(|N_i|-1)w^t_{m\to i}(\ell)(1-\lambda^t)$ and overall contribution is $(|N_i|-1)(1-\lambda^t)\sum_{f_m\in N_i\backslash{f_\ell}}w^t_{m\to i}(\ell)=(|N_i|-1)(1-\lambda^t)$, which is the same as BP if $\lambda^t=0,$ and otherwise, DBP.
>
> Besides, it is noteworthy that new estimation cannot bias the computation, since the old estimate was also scaled by $|N_i|-1$. By the linearity of multiplication, the weights of the new estimation and the old estimation are summed to 1.
>
> **3.  Regarding the description of Toulbar2 in experimental result**
>
> **Response:**  By saying “Toulbar2 performs poorly”, we essentially refer to the fact that Toulbar2 fails to find high-quality solutions even if a very generous runtime budget (i.e., 20min) is given.
>
> **4.  Regarding the experimental results of Toulbar2 and MBE on WGCPs**
>
> **Response:**  Since the domain size in WGCP experiments is set to a relatively small number (i.e., 5), MBE with i-bound of 9 can perform variable elimination quickly. For the performance of Toulbar2 on WGCPs with |$X$|=60, we have double-checked the correctness of the results. The reasons could be that 1) the domain size of each variable is relatively small, making the problems easier to be solved by a search-based solver like Toulbar2, and 2) the constraint functions are highly-structured (i.e., most entries are close to 0), making it easier to get a high-quality upper bound and therefore lead to more efficient pruning. We have included the discussion in line 533-536.
>
> **5. Tuning the damped factor of DBP(-SCFG)**
>
> **Response:**  Please refer to our response 1 to Reviewer jt6S

---

> > ### Comment · Reviewer_rbQp · 2022-08-09
> > **Response**
> >
> > Thank you the authors for answering my comments and for making clear the changes on the paper.
> > I've modified original rating accordingly

---

### Official Review · Reviewer_1zWz · 2022-07-10

**Rating:** 4
**Confidence:** 4
**Soundness:** 2 fair
**Presentation:** 2 fair
**Contribution:** 2 fair

**Summary:**

This paper proposed a deep attentive belief propagation (DABP) model for solving constraint optimization problems (COPs). The proposed DABP generalizes over damping belief propagation by considering dynamic damping factors. In addition, DABP considers different neighbor weights for optimal message composition. Both damping factors and neighbor weights are estimated via neural network units. DABP is trained in a self-supervised manner and the training loss is defined based on a smoothed surrogate of the original objective of COPs. Extensive experiments are performed to demonstrate the effectiveness of the proposal DABP.

**Questions:**

## Theoretical justifications are insufficient
1. It is not clear how dynamic damping factors improve the inference performance theoretically.

2. Though weighted neural messages under a graph neural network framework have been shown to be effective in improving performance by existing works, the messages in belief propagation algorithm are not weighted for aggregation by theory. There is a lack of justification on weighting messages. Furthermore, it is not clear to me why the message weights should vary from iteration to iteration.

3. Is there any theoretical justification on how the inference performance is affected by message weights?

4. Since the proposed smoothed cost can be considered as a relaxation of the original objective of COPs, it would be good if the authors can provide theoretical justifications on how the inference error is introduced by the relaxation.

## Some related works are missing
Existing optimization-based algorithms are developed based on belief propagation algorithm. Messages and beliefs are improved from the variational belief propagation point of view, e.g., the tree-reweighted belief propagation [a] and factional belief propagation [b]. It would be better if the authors can provide a comparison between the proposed method and related works in revising messages / beliefs.

> [a] Wainwright, Martin J., Tommi S. Jaakkola, and Alan S. Willsky. "Tree-reweighted belief propagation algorithms and approximate ML estimation by pseudo-moment matching." International Workshop on Artificial Intelligence and Statistics. PMLR, 2003.

> [b] Wiegerinck, Wim, and Tom Heskes. "Fractional belief propagation." Advances in Neural Information Processing Systems 15 (2002).

## Computational complexity
The training objective (Eq. 14) requires a summation over all $\tau$ and $f$, which would be of high computational complexity. Can the proposed method scale up to large models?

**Minor question:**
1. In line 10, 'massage' should be 'message'.
2. $M_\theta$ in Eq. 7 is not explained.


**Limitations:**

The authors provide discussions on possible limitations of the proposed method.

**Strengths And Weaknesses:**

## Strengths:
**Clarity:** The paper is overall well written. The paper is easy to follow.

**Novelty:** A smoothed cost objective is defined based on COPs. It is an interesting idea to leverage the smoothed cost objective for training without requiring annotations. Leveraging neural networks for improving classical well-established algorithms becomes an attractive topic recently. This paper introduces attention mechanism for improving belief propagation on factor graph, which could be of interest to other related researchers.

## Weaknesses:
**Technical originality:** The technical originality of the paper is marginal. The proposed method is an extension to damping belief propagation. Leveraging attentions to weight messages has been widely explored in different tasks [a].
> [a] Zhang, Li, et al. "Dynamic graph message passing networks." Proceedings of the IEEE/CVF Conference on Computer Vision and Pattern Recognition. 2020.

**Quality:** The quality of the paper can be improved in two aspects: 1) there are missing related works; 2) the proposed method is overall lack of theoretical justifications.

**Significance:** The proposed DABP outperforms SOTA baselines only marginally while the computing time is significantly longer than SOTA baselines. In Table 1, on small-work networks with |X|=100, DABP achieves cost 25.65 which is slightly better than DBP-SCFG (cost = 26.13). However, DABP takes 7m31s while DBP-SCFG only takes 2m10s. The proposed training objective (Eq. 14) seems to be of high computational complexity.

***I detail my comments on the weaknesses in the Question section below.***

---

> ### Author Response · Authors · 2022-08-02
> **Responses to Reviewer 1zWz**
>
> We thank Reviewer 1zWz  for insightful comments and helpful feedback on our work. We address Reviewer 1zWz's suggestions and respond to specific comments below
>
> **1. Regarding the technical originality**
>
> **Response:** The goal of this work is not just to simply leverage attention to weight messages but rather to come up with a novel self-supervised DNN-based BP, DABP, for solving COPs by seamlessly integrating BP, Gated Recurrent Units (GRUs), and Graph Attention Networks (GATs) within the message-passing framework to reason about dynamic weights and damping factors for composing new BP messages.
>
> We want to highlight that, different from the work of Zhang et al. which targets deep representation learning, we do not aim to advance graph representation learning but rather to leverage deep learning to configure BP automatically for solving COPs. Collectively, building on top of GRUs and GATs, we aim to infer the optimal damping factors and message weights of BP by capturing the message dynamics with GRUs and embedding the factor graphs with GATs; Our model is then trained with a novel self-supervised learning loss. Of course, one can replace GATs in our model with a more advanced graph embedding technique such as the method of Zhang et al. to further improve our model.
>
> > Zhang, Li, et al. "Dynamic graph message passing networks." In: CVPR 2020.
>
> **2. Regarding the significance of DABP**
>
> **Response:** Our DABP’s computing time mostly depends on the number of rounds to run (cf. the hyperparameter “R” in line 2 of Algorithm 1), and DABP can be faster than all baselines by setting a smaller R. In our experiments, by setting R=5, DABP always finds better solutions than the strongest baseline DBP-SCFG within comparable computing time (cf. Table 1). In your example, on small-work networks with |X|=100, DABP (R=5) achieves a cost of 25.67 which is better than DBP-SCFG (cost = 26.13), but DABP only takes 1m56s while DBP-SCFG takes 2m10s. Therefore, one can vary R according to computing budgets, and we find that a small R is often enough.
>
> The proposed training objective (Eq. 14) may seem computationally expensive; it can be implemented effectively with GPUs in parallel since each component can be computed independently.
>
> **3. Regarding theoretical justifications**
>
> **Response:** Regarding the theoretical justifications of dynamic damping factors and message weights, these are outside the scope of this piece of research work. Because the theoretical justifications for using damping factors to improve the performance of BP is a challenging open question (cf [c]),  investigating the theoretical justifications of dynamic damping factors and message weights could be even harder.
>
> > [c] Cohen, Liel, Rotem Galiki, and Roie Zivan. "Governing convergence of Max-sum on DCOPs through damping and splitting." Artificial Intelligence 279 (2020): 103212.
>
> We want to highlight that the goal of this work is to propose a deep learning framework to configure and generalize traditional COP solvers such as BP to reduce the human labor to finetune such solvers’ hyperparameters. Intuitively, our model benefits from inferring the optimal damping factors and neighbor weights for each iteration automatically by seamlessly integrating BP and DNNs, and thus, we have more fine-grained control for composing new messages to trigger effective exploration. We have conducted extensive experiments to evaluate the effectiveness of our model.
>
> **4. Regarding the error bound of smoothed cost**
>
> **Response:** We thank the reviewer for the suggestion of analyzing the error bound. We have added a theorem to analyze the error bound induced by smoothed cost (Theorem 1) in Sect. 3.2.
>
> **5. Regarding the missing related work**
>
> **Response:** We thank the reviewer for suggesting the 2 related papers. For the traditional BP-based optimization solvers for COPs, DBP-SCFG [c] is currently the strongest baseline, so we only compare with it.
>
> On the other hand, TRBP [a] and Fractional BP [b] are another class of methods that cope with non-convergence of BP by efficiently solving an LP relaxation of the combinatorial problem, then selecting value assignments based on the solution to the LP. But, they rely on predetermined and static weights, while we automatically learn the best weight through our self-supervised objective, eliminating the need for prior domain knowledge.
>
> We have discussed and cited these 2 related papers in our revised paper.
>
> **6. Regarding computational overhead of the smoothed cost**
>
> **Response:**  For each constraint function, we only need to sum up the combinations of variables in its scope, rather than the combinations of all variables. Besides, by leveraging the powerful vectorized computation offered by modern GPUs, the overhead of computing the smoothed cost can be greatly reduced. Therefore, our DABP can scale up to large instances. Please also refer to our response 5 to Reviewer DKJk.

---

> > ### Comment · Reviewer_1zWz · 2022-08-07
> > **Response**
> >
> > Thanks for the authors’ responses. All my concerns are responded, and I appreciate the efforts the authors spend in answering my questions. Unfortunately, my major concerns are not fully addressed by the responses.
> >
> > My major concern still lies in the technical originality and theoretical justifications. The authors claim that the technical originality lies in proposing a self-supervised DNN-based BP whereby deep learning is leveraged to configure BP automatically. However, neural networks are employed more in a heuristic way without sufficient theoretical justifications (or at least some theoretical intuitions) on how the proposed damping factors and dynamic weights can improve upon vanilla BP.
> >
> > In addition, it is important to contrast the proposed method with existing BP algorithms systematically to highlight the motivation and importance of the proposed damping factors and dynamic weights.  Only comparing to two algorithms (TRBP and Fractional BP) as provided in the response is not sufficient.
> >
> > Based on my justifications above, I will keep my original rating.

---

> > > ### Author Response · Authors · 2022-08-08
> > > **Further clarifications**
> > >
> > > We sincerely thank the reviewer for the further comments. We realize that it is very challenging to give rigorous theoretical analysis of our DABP due to the inherent complexity induced by dynamic hyperparameters and neural networks [1, 2, 3], and we have instead demonstrated the effectiveness of our DABP through extensive experiments in our paper, where we have already noted in our previous responses. **However, we still want to try our best to address your concerns by giving some high-level intuitions about our method from the learning perspective as follows.**
> > >
> > > Since our model is self-supervised by the smoothed cost which can serve as a surrogate for the true solution cost in each BP iteration, the negative gradients of the smoothed cost w.r.t. probability distribution reflects the improving direction (cf. E	q. (6)). Such gradients are propagated backward through the chain rule to the beliefs, BP messages, **neighbor weights** and **damping factors** and finally, our DABP model parameters. Therefore, by adjusting the parameters of our model according to the improving direction through a suitable optimizer (e.g., Adam), we can improve neighbor weights and damping factors, alter the composition strategy of BP messages, and hence result in beliefs that induce a better solution (and smoothed cost).
> > >
> > > Regarding the related work, **we have further surveyed comprehensively and compared our method with a wide range of related techniques for improving loopy BP**, including: (1) traditional methods like relaxation, alternatively directed message-passing, damping, etc.; (2) LP-based methods like MPLP, TRBP, Fractional BP and Norm-Product and (3) neural-based methods that learn to modify the BP messages, e.g., BPNN, FGNN and NEBP. Please kindly refer to the updated related work section of our paper.
> > >
> > > > [1]  Cohen, Liel, Rotem Galiki, and Roie Zivan. "Governing convergence of Max-sum on DCOPs through damping and splitting." Artificial Intelligence 279 (2020): 103212.
> > >
> > >  >[2] Jonathan Kuck, Shuvam Chakraborty, Hao Tang, Rachel Luo, Jiaming Song, Ashish Sabharwal, and Stefano Ermon. Belief propagation neural networks. In NeurIPS, pages 667–678, 2020.
> > >
> > > > [3] Victor Garcia Satorras and Max Welling. Neural enhanced belief propagation on factor graphs. In AISTATS, pages 685–693, 2021.

---

> > > > ### Comment · Reviewer_1zWz · 2022-08-08
> > > > **Response**
> > > >
> > > > Thanks for the further clarifications. I appreciate the efforts in providing a comprehensive review of related works to address my concerns. As for theoretical justification, I agree that it could be challenging to provide theoretical analysis of the proposed algorithm. But it is important to justify why and how dynamic damping factors and message weights can improve the performance and to motive the usage of neural networks. I would take the authors' further clarifications into my consideration to reach my final assessment.

---

### Official Review · Reviewer_DKJk · 2022-07-11

**Rating:** 6
**Confidence:** 4
**Soundness:** 3 good
**Presentation:** 3 good
**Contribution:** 3 good

**Summary:**

The paper proposes a new belief propagation algorithm for COPs (constraint optimization problems), called DABP (deep attentive BP). DABP increases the granularity of the damping factors and neighbor weights by allowing them to be dynamic and specific for every variable node. The dynamic damping factors and neighbor weights are automatically inferred for each iteration by DNN, using GRUs (gated recurrent units) and GATs (graph attention network) and a multi-head attention layer. DABP uses novel smooth self-supervised learning loss, therefore avoiding the need for expensive training labels. The experimental evaluation shows that DABP achieves a high convergence rate and performs better than some SOTA baselines.

**Questions:**

1) It would be useful to include the memory consumption of the algorithms. In particular, how much memory does DABP use, main memory and GPU memory? Memory availability can impact the performance of some of the algorithms.

- why did you limit MB i-bound to 9? Am I correct to assume that domain size 15, and i-bound 9 reaches the main memory limit?
Just as a reference, for a more fair comparison, there are MB implementations that exploit GPU, e.g.:
F. Bistaffa, N. Bombieri and A. Farinelli, "An Efficient Approach for Accelerating Bucket Elimination on GPUs," in IEEE Transactions on Cybernetics, vol. 47, no. 11, pp. 3967-3979

- My understanding is that Toulbar2 is a search based algorithm. Is Toulbar2 using caching during search? Do you know its memory usage?

2) How many GPUs do you use in your setup? Do they have 24GB memory each?

3) The networks are relatively small. How does DABP scale for bigger networks?

**Ethics Review Area:**

["I don’t know"]

**Strengths And Weaknesses:**

Strengths:
The paper combines and refines several existing schemes, while adding novelty items. The damping scheme is refined as much as possible by allowing node specific values for damping factors and neighbor weights. These values are learned for each instance, and the use of GRUs prevents the gradient vanishing issue. A smoothed cost is proposed as a surrogate objective, allowing the learning to happen in the absence of labels.

The presentation is clear, considering the need for relatively involved notation. The Figure 2 and the Algorithm 1 description are very useful.

Weaknesses:
The significance of the proposal depends a lot on the experimental evaluation, which is quite extensive, but could benefit from some more clarifications, see questions. The size of the networks is still small, how far can DABP scale?

---

> ### Author Response · Authors · 2022-08-02
> **Responses to Reviewer DKJk**
>
> We thank Reviewer DKJk for insightful comments and helpful feedback on our work. We address Reviewer DKJk’s suggestions and respond to specific comments below.
>
> **1. Regarding the memory footprints**
>
> **Response:** We thank the reviewer for the suggestion of including CPU and GPU memory footprints. We have included the GPU memory footprint in Appendix C.6. Besides, DABP has a similar CPU memory footprint of 4.3GB on all test cases. Therefore, we choose to not report the CPU memory footprint separately.
>
> **2. Regarding the memory bound of MBE**
>
> **Response:** We thank the reviewer for the question and for suggesting the GPU implementation of bucket elimination. We set the memory bound based on our computational resources. We have made this point clear in lines 496-497. For GPU implementation, we would like to note that the bottleneck of solution-quality performance in MBE mainly depends on the available memory which directly determines the maximum tables it can process, rather than intractable runtime. Besides, the code provided by the author of [1] is no longer available, which hinders the direct comparison with [1].
> > [1]Bistaffa, Filippo, Nicola Bombieri, and Alessandro Farinelli. An efficient approach for accelerating bucket elimination on GPUs. IEEE Transactions on Cybernetics 47.11 (2016): 3967-3979.
>
> **3. Regarding the caching and memory consumption of Toulbar2**
>
> **Response:** We set HBFS [2] as the tree-search algorithm for Toulbar2, which does not use caching during the solving process. We have also tested its BTD counterpart (i.e., BTD-HBFS) which performs caching during the search, but the performance is worse than HBFS.
> > **[2] D Allouche, S de Givry, G Katsirelos, T Schiex and M Zytnicki. Anytime Hybrid Best-First Search with Tree Decomposition for Weighted CSP. In CP, pages 12-28, 2015.**
>
> The following table presents the memory usage of Toulbar2 (in MB).
>
> | 　        | Random COPs | WGCPs | SF Nets | SW Nets |
> |-----------|-------------|-------|---------|---------|
> | \|$X$\|=60  | 28.69       | 30.33 | 31.02   | 31.28   |
> | \|$X$\|=80  | 50.36       | 55.26 | 36.91   | 36.37   |
> | \|$X$\|=100 | 55.47       | 56.43 | 51.48   | 46.24   |
>
> **4. Regarding the number of GPUs used in the experiments**
>
> **Response:** In all our experiments, we only use one GPU with 24GB of memory.
>
> **5. Regarding the scalability of DABP**
>
> **Response:** To test the scalability limit of our DABP, we generate random COPs with a graph density of 0.05 and a step size of 50 variables, starting the problems with 150 variables. The results show that our algorithm can scale up to the problems with 300 variables. The following table presents the GPU memory footprint (in GB) for each experiment.
> | \|$X$\|  | 150  | 200   | 250   | 300   |
> |----------|------|-------|-------|-------|
> | GPU memory footprint | 5.86 | 10.23 | 16.58 | 21.94 |

---

> > ### Comment · Reviewer_DKJk · 2022-08-10
> > **Thank you**
> >
> > Thank you for the very detailed and helpful response. I don't have any more questions now.

---

### Official Review · Reviewer_jt6S · 2022-07-14

**Rating:** 6
**Confidence:** 4
**Soundness:** 2 fair
**Presentation:** 3 good
**Contribution:** 2 fair

**Summary:**

The authors augment the max-sum algorithm with learning in order to solve COPs. The augmentation involves the used GRUs and GATs to produce an alternative, heavily parametric version of max-sum. The cost function can be expressed in terms of one-hot encodings of the solutions for each variable. A softened version of this cost is obtained by using the soft-max of the computed beliefs instead. The parameters of the GRUs and GATs can now be chosen to optimize this softened cost, thus directly using the cost to drive the learning, and not some golden target beliefs.

**Questions:**

1. Can you clarify if a single trained DABP is used in _all_ the experiments, or if a different one is used for each model type (or other approach)? Also, how many examples is it trained on? Or is it "trained" separately on each new instance to solve? All this wasn't clear to me from reading the paper.

2. The provided time for DABP is the time to run inference only, or does it include the "learning" time?

3. Why not test DBP with other damping parameters?

4. Between DBP (essentially one parameter) and DABP (many many parameters), there seems to be a world of "slightly flexible" DBPs. Given the slim advantage of DABP over DBP, is DABP really the minimum amount of sophistication necessary to improve on DBP?

Minor comment:
- nature choice -> natural choice

**Limitations:**

Described above.

**Strengths And Weaknesses:**

Strengths:

- Technically correct
- Clear presentation
- Interesting problem
- Does not rely on training data with correct solutions; thus potential to solve COPs that aren't solvable by other methods

Weaknesses:

- The competitor DBP is not pushed hard enough, for instance by exploring more damping factors
- The small advantage obtained wrt DBP doesn't seem to justify all the additional machinery deployed. Also, DBP doesn't require training, can be deployed directly for any new architecture.

---

> ### Author Response · Authors · 2022-08-02
> **Responses to Reviewer jt6S**
>
> We thank Reviewer  jt6S for insightful comments and helpful feedback on our work. We address Reviewer jt6S’s suggestions and respond to specific comments below.
>
> **1. Tuning the damped factor of DBP(-SCFG)**
>
> **Response:** We thank the reviewer for the suggestion of tuning the damping factors. In our experiments, we set the damping factor to 0.9 for DBP(-SCFG) according to the recommendation of [1], since we use the similar benchmark problems (e.g., random COPs, scale-free, etc.). We have varied the damping factor from 0.5 to 0.9 with a step size of 0.1, and the results can be found in Appendix C.3 (see Fig. 5-7). It can be observed that our DABP still outperforms DBP and DBP-SCFG, while DBP and DBP-SCFG require significantly longer runtime if we tune the damping factor (see Fig. 8).
>
> > [1] Liel Cohen, Rotem Galiki, and Roie Zivan. Governing convergence of max-sum on DCOPs through damping and splitting. Artificial Intelligence, 279:103212, 2020
>
>
> **2. The significance of DABP**
>
> **Response:** To the best of our knowledge, DBP-SCFG is currently the strongest approximate solver for large-scale COPs. Therefore, improving DBP(-SCFG) by 1.46%-27.5% without incur much additional runtime overhead is a non-trivial advance over SotA. In fact, if we use 5 times of restart, our DABP’s runtime performance is comparable to (or even better than) the one of DBP(-SCFG). Finally, our DABP has a nice anytime property, which can return the best solution found so far within the user-specified runtime budget.
>
> **3. Regarding the model for solving COPs**
>
> **Response:** For each instance to be solved, we always train the model from scratch, i.e., start from an untrained model with initial weights, and continuously improve it on the current instance via online-learning. In other words, there is no pretraining phase and therefore no training example is required. We have made this point clear in row 174-176.
>
> **4. Regarding the reported runtime**
>
> **Response:** The reported runtime for DABP covers all the subprocedures, including learning, inference and BP message-passing.
>
> **5. Regarding the complexity and "slightly flexible" DBPs**
>
> **Response:** The parameters of our DABP comprise of $O(T|X|d^2)$ heterogenous neighbor weights and $O(T|X|d)$ heterogenous damping factors.  Therefore, to demonstrate the necessity of the parameters, we conduct ablation study by assuming the equal neighbor weights (which is the major source of complexity) and reasoning only about $O(T|X|d)$ heterogenous damping factors, or one average damping factor.
>
> The results can be found in Appendix C.5. It can be observed that the performance degenerates if we do not reason about neighbor weights, and the gap is widened if we only reason about one average damping factor, which highlights the necessity of our proposed heterogenous damping factors and neighbor weights.

---

### Meta-Review · Area_Chair_5o76 · 2022-08-25

**Recommendation:** Accept
**Confidence:** Less certain

**Metareview:**

4 knowledgable reviewers reviewed the paper, 3 of them recommending weak acceptance, 1 borderline rejection. The reviewers engaged with the authors and a discussion among the reviewers took place. The reviewers appreciate the considered problem, the novelty of the proposed approach and the reported performance improvements. At the same time, there are concerns regading the theoretical justifcation of the method, relation to existing work, and comparison with other existing methods (lacking baselines and pushing the baselines to the limit). There was a discussion regading the need for a theoretical justification and I side more with the reviewers which argue that such a justification is not absolutely necessary -- nevertheless, more motivation and intutition about the proposed approach should still be provided. In summary, the paper is viewerd borderline, which I agree with, but I think there are some relevant contributions which could be interesting to the community. Hence I am recommending acceptance of the paper but strongly encourage the authors to carefully consider all comments and suggestions which came up in the reviews and discussions with the reviewers when preparing the final version of their paper.

**Award:**

No

---

### Decision · Program_Chairs · 2022-09-14

Accept